# A Sensor-Based Feedback Device Stimulating Daily Life Upper Extremity Activity in Stroke Patients: A Feasibility Study

**DOI:** 10.3390/s23135868

**Published:** 2023-06-25

**Authors:** Anthonia J. Langerak, Gerrit Ruben Hendrik Regterschot, Marc Evers, Bert-Jan F. van Beijnum, Carel G. M. Meskers, Ruud W. Selles, Gerard M. Ribbers, Johannes B. J. Bussmann

**Affiliations:** 1Department of Rehabilitation Medicine, Erasmus MC, University Medical Center Rotterdam, 3000 CA Rotterdam, The Netherlands; a.j.langerak.1@erasmusmc.nl (A.J.L.);; 2Rijndam Rehabilitation, 3015 LJ Rotterdam, The Netherlands; 3Department of Biomedical Signals and Systems, University of Twente, 7522 NB Enschede, The Netherlands; 4Department of Rehabilitation Medicine, Amsterdam Neuroscience and Amsterdam Movement Sciences, Amsterdam UMC, Vrije Universiteit, 1081 HV Amsterdam, The Netherlands; 5Department of Plastic and Reconstructive Surgery, Erasmus MC, University Medical Center Rotterdam, 3000 CA Rotterdam, The Netherlands

**Keywords:** stroke, rehabilitation, accelerometry, upper extremity, daily life, arm usage, feedback

## Abstract

This study aims to evaluate the feasibility and explore the efficacy of the Arm Activity Tracker (AAT). The AAT is a device based on wrist-worn accelerometers that provides visual and tactile feedback to stimulate daily life upper extremity (UE) activity in stroke patients. Methods: A randomised, crossover within-subject study was conducted in sub-acute stroke patients admitted to a rehabilitation centre. Feasibility encompassed (1) adherence: the dropout rate and the number of participants with insufficient AAT data collection; (2) acceptance: the technology acceptance model (range: 7–112) and (3) usability: the system usability scale (range: 0–100). A two-way ANOVA was used to estimate the difference between the baseline, intervention and control conditions for (1) paretic UE activity and (2) UE activity ratio. Results: Seventeen stroke patients were included. A 29% dropout rate was observed, and two participants had insufficient data collection. Participants who adhered to the study reported good acceptance (median (IQR): 94 (77–111)) and usability (median (IQR): 77.5 (75–78.5)-). We found small to medium effect sizes favouring the intervention condition for paretic UE activity (η^2^G = 0.07, *p* = 0.04) and ratio (η^2^G = 0.11, *p* = 0.22). Conclusion: Participants who adhered to the study showed good acceptance and usability of the AAT and increased paretic UE activity. Dropouts should be further evaluated, and a sufficiently powered trial should be performed to analyse efficacy.

## 1. Introduction

Strokes are a significant cause of disability worldwide [1]. Over two-thirds of stroke patients experience upper extremity (UE) motor impairments, resulting in a decreased capacity to execute a task or action in a standardised environment and reduced performance of activities in their natural environment [2]. Through high-intensity exercise therapy, UE capacity can improve [3,4]. However, there is no one-on-one relation between the level of UE capacity and the performance of UE activities in daily life [5,6]. This can be explained by patients’ tendency to compensate for their contralesional paretic UE with their ipsilesional non-paretic UE, even when paretic UE capacity has increased. This leads to a small amount and low intensity of paretic UE use in daily life, the so-called ‘non-use’ phenomenon [7,8].

A renowned high-intensity exercise therapy to improve UE activities and prevent or treat non-use by stimulating UE use in daily life is constraint-induced movement therapy (CIMT) [9]. During CIMT, patients perform highly intensive supervised UE therapy and are forced to use the paretic UE by immobilising the non-paretic UE for up to 90 per cent of the day. CIMT can improve UE capacity and self-reported UE use in daily life in stroke patients, and those improvements are sustained over time [10]. However, CIMT requires a high level of patient motivation, which is a common challenge for stroke patients (11). It often involves intensive supervision, which induces high costs [10,11]. Furthermore, CIMT does not allow bi-manual use of the UE, although most UE activities are bimanual in nature [12]. Therefore, there is an urgent need for an intervention that stimulates paretic UE use in daily life without the disadvantages of CIMT.

Wearable activity trackers are well-established tools for real-time UE activity tracking in stroke patients [13]. Activity trackers can be used safely outside the clinical setting, can measure both arms simultaneously, and can objectively provide insight into the quantity of daily life UE use of an individual patient in real-world settings [14]. Moreover, wearable activity trackers can provide real-time, personalised feedback on UE activity [15]. Real-time feedback, based on objective measurement, has the potential to effectively stimulate physical activity in people with various conditions, including stroke [16]. Although several devices are available to measure stroke patients’ UE use in daily life, activity trackers providing valid measurements and personalised feedback on daily life UE use in stroke patients are not yet widely available in clinical practice [15,17].

Therefore, we developed the Arm Activity Tracker (AAT) based on a user-centred approach involving stroke patients, therapists, and clinicians. The AAT is based on a previously developed activity detection device [5,18] and consists of accelerometer wristwatches on both wrists and an accelerometer on the non-paretic thigh. This system validly measures paretic UE activity during sitting and standing in daily life [18]. Based on patient-specific goals for paretic UE activity, which were pre-defined by the patient and their physiotherapist, the AAT on the paretic UE provides visual feedback and vibrotactile stimuli. In this way, the AAT motivates and reminds patients instead of forcing them to use the paretic UE.

This study aimed to evaluate the feasibility of the AAT in three domains: (1) treatment adherence, (2) user acceptance and, (3) usability of the AAT when integrated into conventional UE rehabilitation treatment in stroke patients. Secondarily, we explored the efficacy of the AAT for improving paretic UE activity and UE activity ratio during the treatment period.

## 2. Materials and Methods

### 2.1. Design

A randomised, crossover within-subject design was used. Participants wore the AAT for five consecutive weeks: one measurement week to determine UE activity level at the baseline, followed by a two-week control and a two-week intervention condition in a balanced and random order across participants. No washout period was included.

### 2.2. Participants

Based on ambiguous examples in the literature about sample size justification for feasibility studies and similar work, we aimed to recruit a minimum number of 12 [19,20] and a preferred number of 20 participants [21] from Rijndam Rehabilitation Centre (Rotterdam, the Netherlands) during inpatient rehabilitation. From April to December 2021, patients at least one week and at most six months after ischemic or haemorrhagic unilateral stroke were screened for eligibility. Patients were screened for eligibility by a physiotherapist based on the following inclusion criteria: (1) eighteen years or older; (2) reduced UE function resulting from stroke as defined by the National Institutes of Health StrokeScale score; (3) able to lift the paretic UE against gravity (>30 degrees shoulder anteflexion); (4) able to attach the devices independently or with the assistance of a caregiver. Exclusion criteria were the presence of (1) UE comorbidities interfering with daily life UE function (e.g., frozen shoulder, severe UE pain); (2) major cardiopulmonary disease; (3) major depression interfering with daily life functions; (4) cognitive impairments or comprehensive aphasia resulting in the inability to provide informed consent.

### 2.3. Ethics

The Medical Ethics Committee of Erasmus Medical Center Rotterdam, the Netherlands, approved the study (MEC-2020-0007), and all participants provided written informed consent. The Arm Activity Tracker used in this study was manufactured by 2M Engineering—Wearable & Medical Devices (Valkenswaard, The Netherlands).

### 2.4. Procedure

#### Randomisation

One researcher (G.R.H.R.) generated a balanced, random allocation sequence for the order of the intervention with a 1:1 ratio for twenty participants using a custom-made script in RStudio (version 1.4.1106, RStudio, Inc., Boston, MA, USA). Randomisation was concealed from the physiotherapist responsible for including participants.

### 2.5. Intervention

The same physiotherapist, who was trained to use the AAT, performed in-person consultations at Rijndam Rehabilitation Center with the participants twice a week during the five-week study period. At the start of the study, participants were instructed to wear the AAT daily, from morning to evening bedtime. During the first-week baseline, UE activity was measured by the AAT without providing feedback to the AAT or therapists. During the two-week intervention, participants received real-time feedback from the AAT on paretic UE activity and the ratio between paretic and non-paretic UE activity (Figure 1B). Participants could instantly request an overview of their daily progress in achieving their goals on paretic UE activity and ratio on the AAT. During the twice-weekly consultations, the physiotherapist set the UE activity goals and feedback frequency using the AAT PC application. The physiotherapist also evaluated paretic UE activity, ratio, and goal achievement with the participant. During the two-week control period, participants wore the AAT without receiving feedback from the device or the physiotherapist.

### 2.6. Device

#### 2.6.1. Hardware

The AAT consists of three small, lightweight accelerometer sensors (Figure 1A): one worn on each wrist and one attached to the front of the non-paretic thigh [22]. The AAT accurately measures UE activity (overall agreement: 75%) by recording UE activity only during sitting and standing periods to correct involuntary UE movement caused by whole-body movements such as walking [18,22]. The accelerometer on the thigh accurately detects whole-body movements and postures in stroke patients such as walking, sitting, and standing (agreement range: 82–98%) [23]. The wrist sensor display on the paretic UE shows real-time feedback on daily UE activity using vibrotactile triggers and visual messages (Figure 1B). Real-time feedback is given according to pre-set goals at pre-set time intervals and only when the paretic UE is not moving (e.g., if patients do not use the paretic UE sufficiently, the system briefly vibrates and shows a blinking visualisation of a hand on the display to remind the patient to use the paretic UE). In addition, summaries of daily progress related to the pre-set goals can be shown upon request using buttons on the AAT (Figure 1B). The wrist sensors had to be charged every night and taken off during water activities such as showering or swimming. The leg sensor is water-resistant and needs to be charged twice a week.

#### 2.6.2. Software

The PC application allows therapists to set and adjust UE activity goals and feedback frequency, detect changes in UE activity over time, and provide personalised coaching to patients. Summary data from the AAT is transferred to the PC application by connecting the AAT to a PC. The PC application visualises UE activity data hourly and daily.

### 2.7. Data Processing

Each accelerometer measures raw acceleration data with a sampling frequency of 12.5 Hz. Per axis of the accelerometer, the difference from the previous sample was calculated for each sample. Next, the magnitude of the axis differences was calculated per sample. Subsequently, a moving average filter was applied every two seconds to obtain the average magnitude across 25 samples. The outcome is energy, where 1024 energy = 1 G = 9.81 m/s^2^. The device stores the data in epochs of twenty seconds. The sensor on the thigh converted acceleration data into movement counts and body postures/movements with a 1.6 Hz resolution [13]. The recognition of body postures and movements (i.e., laying/sitting, standing, and walking) by the sensor on the thigh was based on (1) the orientation of the sensor compared to gravity and (2) the intensity of the movement (in movement counts) [18]. An Activ8 sensor on the thigh accurately recognises whole-body postures and movements in stroke patients with an accuracy ranging from 82 to 100 per cent [23].

### 2.8. Outcomes and Measures

#### 2.8.1. Baseline Characteristics

The physiotherapist or a trained research assistant collected baseline data within five days before a participant started the study. Valid and reliable measures were used to assess UE function (Fugl-Meyer upper extremity score; FM-UE) [24], UE capacity (Action Research Arm Test; ARAT) [24], stroke severity (National Institutes of Health Stroke Scale; NIHSS) [25], and affinity with technology (Affinity Technology Inventory; ATI) [26]. Sociodemographic data were collected from electronic patient records.

#### 2.8.2. Primary Outcomes

The main feasibility-related study parameters were adherence, patient-reported acceptance and usability of the AAT.

Adherence was defined as (1) the dropout rate during the study period; (2) the rate of insufficient AAT data collection (activity data available less than 8 h a day for at least four days per week). The trained physiotherapist asked and reported participants’ reasons for dropping out during the baseline, intervention and control conditions.

Acceptance of the AAT was measured using the technology acceptance model questionnaire (TAM). The TAM questionnaire evaluates perceived usefulness, ease of use, and technology acceptance [27]. It contains sixteen items, scored on a seven-point Likert scale, where seven indicates ‘strongly agree’. In a healthy English-speaking population, the TAM has shown sufficient validity and reliability [28]. For the Dutch-speaking participants in this study, we translated the TAM into Dutch. The TAM has been used in Dutch-speaking populations before [29].

In addition, adverse events reported spontaneously by the participant or observed by the physiotherapist were documented. Adverse events were defined as any undesirable experience occurring to a participant during the study, whether or not it was considered related to the AAT.

The usability of the AAT was measured using the system usability scale (SUS). The SUS is a ten-item questionnaire that evaluates perceived usability on a five-point Likert scale, where five indicates ‘strongly agree’ [30]. Scores range from zero to 100, where 100 represents optimal usability. An SUS score of above 68 can be considered adequate usability [31]. The SUS is valid and reliable in Dutch [32]. In addition, technical and operational issues with the AAT or the PC application reported by the participant and the physiotherapist were documented.

#### 2.8.3. Secondary Outcomes

The outcome measures of the AAT were:(1)The amount of activity of the paretic UE, which was measured as the average amount of energy per sitting and standing hour during baseline, control and intervention periods.(2)The UE activity ratio, which was measured as the average amount of energy per sitting and standing hour of the paretic UE divided by the average amount of energy per sitting and standing hour of the non-paretic UE during baseline, control and intervention periods.

### 2.9. Statistical Analysis

All data were analysed in RStudio (version 1.4.1106, RStudio, Inc.). We summarised the baseline characteristics of participants overall and separately for participants receiving the intervention or the control condition first, with mean and standard deviation (SD) or median and interquartile range (IQR) for continuous variables (as appropriate) and number and per cent (%) of the total for categorical variables.

We analysed the dropout rate as the number of participants who withdrew from the study divided by the number of participants included. For all further analysis, we used the available cases. The rate of insufficient data collection was analysed as the number of participants with insufficient data divided by the number of participants included.

Total, median and IQR scores were calculated per item for SUS and TAM. Concerning the secondary outcome, the paretic UE activity, non-paretic UE activity and UE activity ratio were analysed for the baseline, intervention and control conditions. Participants with insufficient data collection were excluded from this analysis. Data were checked on normality (QQplots, Shapiro test) and outliers (box plots, median, IQR; where values above Q3 + 1.5 × IQR or below Q1 − 1.5 × IQR were considered outliers).

The mean energy for both variables was calculated per hour for each participant in baseline, intervention and control conditions. Subsequently, mean and standard deviation (SD) were calculated per participant and on condition level for each outcome variable.

We conducted a two-way mixed ANOVA type = 3 for unequal group sizes (R package ‘ezANOVA’) to determine the effect of the intervention on UE activity (within subjects) and the effect of the order of the intervention (between subjects). Assumptions of homogeneity of variances, covariances and sphericity were checked (resp. Levene’s tests, mbox test, Mauchly’s test). A *p*-value below 0.05 was considered statistically significant. For significant effects from the ANOVA, we performed post hoc pairwise T-tests with a Bonferroni correction to correct for multiple testing. Generalised eta squared (η^2^_G_) with confidence intervals (CI) were calculated to identify the effect sizes of the intervention. Effect sizes were deemed small (0.02), medium (0.13), or large (0.26) [33]. In the case of outliers, we ran the analysis with and without outliers included to investigate the influence of those outliers. 

## 3. Results

### 3.1. Participants

Twenty-three patients were assessed for study eligibility, six of whom refused to participate. In total, seventeen participants were included. Due to COVID-19 restrictions, we could not continue the inclusion of the up to twenty participants we aimed for. After the one-week baseline, ten participants received the intervention condition first, and seven received the control condition first. Participant characteristics are reported in Table 1. Participants started with the study at a median of 33 (range: 25–60) days after stroke and had a heterogeneous UE function (FMA UE median: 43, IQR: 35–54) and capacity (ARAT median: 24, IQR: 13–45) at baseline.

### 3.2. Primary Outcomes

#### 3.2.1. Adherence

Five out of seventeen participants (29.4%) dropped out of the study. During the baseline measurement period of one week, one participant withdrew due to a nickel allergy to the straps of the device and one withdrew due to the high psychological burden of wearing the device in combination with other rehabilitation therapies. During the intervention period, three participants withdrew: two due to the high psychological burden of wearing the device in combination with other rehabilitation therapies and one because of an unexpected early discharge. Two out of the twelve participants (16.7%) who completed the study protocol had insufficient data collection due to technical issues with the device; the devices shut down and did not collect data when being worn by the participants. Both participants were excluded from data analysis on secondary outcomes.

#### 3.2.2. Acceptance

The twelve participants who completed the study protocol reported good acceptance on the TAM questionnaire (median 94.0; IQR 77.0–111.0). Participants rated items 1, 2 and 3 regarding usability highest (median 7, IQR 6–7). The lowest score was given for item 11 (median 3.5, IQR 1.5–6), indicating that participants’ willingness to use the device at home was medium and differed between participants. Per item and total scores are shown in Table 2. One adverse event was reported during the study: a nickel allergy to the straps of the device.

#### 3.2.3. Usability

The twelve participants who completed the study protocol scored above the pre-set score of 68 on the SUS (median 77.5; IQR 75.0–87.5), indicating adequate usability [31]. Items 2, 3, 4, 8, and 10 were scored most extremely (for items 2, 4, 8, 10: median 1; IQR 1–2 and for item 3: median 4.5; IQR 4–5), indicating that the participants strongly disagree with the system being complex and strongly agree with the device being easy to use. Per item and total scores are shown in Table 3.

### 3.3. Secondary Outcomes

Paretic UE activity and UE activity ratio were analysed in ten participants. Paretic and non-paretic UE activity and ratio during baseline, intervention and control periods are presented on group and individual levels (Figure 2A–C). The figures show increased paretic UE activity and ratio during the intervention compared to baseline and control conditions, with substantial differences in individual participant responses during the three conditions.

Data did not significantly differ from a normal distribution (Shapiro test > 0.05). For the UE activity ratio, two outliers were identified. These were manually checked and could be explained by the level of UE functioning of those patients and therefore remained in the analysis. Other ANOVA assumptions were met. The analysis showed an effect of the conditions (F = 7.39, *p* = 0.005) and no effect of the intervention order (F = 0.94, *p* = 0.361) on total daily paretic UE activity. Post hoc tests showed an increase in paretic UE activity during the intervention period compared to the baseline (+12.9%, *p* = 0.005) and compared to the control condition (+14.2%, *p* = 0.041). We found a small to medium effect size of the intervention (η^2^_G_ = 0.07) on paretic UE activity compared to the control period. On the UE activity ratio, we also observed an effect of the conditions (F = 9.1, *p* = 0.002) and an effect of the intervention order (F = 8.34, *p* = 0.025). Post hoc tests showed an increase in UE activity ratio during the intervention period compared to the baseline (+51.24%, *p* = 0.005) but not significantly compared to the control condition (+22.05%, *p* = 0.218). We found a large effect size of the intervention (η^2^_G_ = 0.26) on the UE activity ratio compared to the control condition. Additional analysis without outliers showed smaller, non-significant differences between intervention and control conditions (+5.89%, *p* = 0.184) and a small effect size (η^2^_G_ = 0.11).

## 4. Discussion

This randomised crossover study evaluated the Arm Activity Tracker’s feasibility (adherence, acceptance and usability) and explored the efficacy of stimulating paretic UE activity in stroke patients. Seventeen participants were included; five of them dropped out, and two had insufficient data collection with the AAT caused by technical issues with the devices. In participants who adhered to the study, the AAT showed good acceptance and usability. Moreover, we found promising efficacy and increased UE activity compared to the control and baseline conditions.

Although the literature shows that dropout rates largely differ between studies in stroke rehabilitation (from 0–83 per cent) and are related to trial characteristics such as trial size, the continent of recruitment and recruitment strategy [34], this study’s 29 per cent dropout rate was higher than what we expected based on a study evaluating the feasibility of a similar device, the CueS wristband. The latter study showed a dropout rate of 12 per cent during a four-week study period [35]. In one case, the reason for dropping out of our study, unexpected early discharge, seemed unrelated to using the AAT. In one case, the participant dropped out due to a nickel allergy. In the future, this could be prevented by using nickel-free straps. In three out of the five dropouts, the reason was the psychological burden of wearing the AAT in combination with other treatments participants received in the rehabilitation centre. The literature indicates that up to 90 per cent of stroke patients reported considerable self-perceived physical, emotional and economic burden post stroke, associated with age, financial pressure, comorbidity and functional status [36]. However, self-perceived burden generally declined within the first three months post stroke [36]. Before planning an efficacy trial with the AAT, the psychological burden of wearing the AAT should be further investigated among these patients and factors associated with self-perceived burden and the timing of the intervention should be carefully considered. Based on these investigations, further optimisation of the AAT’s design should be discussed with the manufacturer. Furthermore, the device failure that led to insufficient data from two participants should be investigated and prevented by optimising the device.

In patients who completed the study, we observed good acceptance and usability of the AAT. Those participants scored above 68 on the SUS, indicating that the system was easy to use. Our results on usability are similar to those found in a study investigating the usability of a similar device, which aimed to improve daily UE activity in stroke patients by stimulating paretic UE activity with vibrotactile triggers [20]. This study showed an SUS score above 70 in 9 out of 10 participants after cross-sectional use of the device [20].

Although our study was not powered to analyse intervention effects, we found promising results regarding the efficacy of the AAT on paretic UE activity and UE activity ratio. We found small to moderate effect sizes for absolute UE activity (η^2^_G_ = 0.07) and ratio (η^2^_G_ = 0.11). Paretic UE activity significantly increased by 14 per cent during the intervention compared to the control condition. This aligns with research in the literature showing an increase in stroke patients’ UE activity of 11–29 per cent after using a vibrotactile feedback watch for four weeks compared to baseline [37]. CIMT literature also shows small to moderate effect sizes, depending on the intervention protocol and the characteristics of stroke patients in the study [10]. Our study sample was heterogeneous regarding UE function, capacity and time post stroke. This might be a reason for the differences between individual participant trajectories in this study. Based on the literature, we hypothesise that the effect size could increase with a longer intervention duration when stimulating UE activity is combined with high-intensity exercises for the UE and when starting in the early sub-acute phase post stroke [10].

The two aforementioned comparable interventions are still under research, and although the results from pilot studies are also promising, their efficacy has yet to be published [20,35,37,38]. Both devices aim to improve daily UE activity in stroke patients by stimulating UE activity using vibrotactile triggers on wrist-worn accelerometers [35,38], but there are differences with the AAT. Neither of the devices mentioned show visual feedback on the activity trackers, which might increase the effects compared to vibrotactile feedback alone. Another advantage of the AAT compared to those systems is the correction of involuntary movements of both UEs due to whole-body movements such as walking. The literature has shown that UE activity is overestimated by 8–41 per cent if data were not corrected for whole-body movements [18]. Therefore, the AAT validly measures UE activity on both arms, body posture and motion detection, and uses personalised vibrotactile and visual triggers to stimulate paretic UE activity.

This study has some limitations. The intervention period in the study was two weeks, which is short compared to most interventions to change stroke patients’ movement behaviour [16] or UE capacity [10]. As we suggested, the duration of the intervention period could have influenced the results found in this study, e.g., the dropout rate and the effect sizes. However, our feasibility study was pragmatically designed to evaluate whether an intervention study could be done instead of performing a miniature version of a main trial [39]. The importance of such work has been recognised as an essential step before conducting clinical trials with large numbers of participants and a longer follow-up [39]. Therefore, our study design, including a shorter intervention period, fits the aims of a feasibility study and provides relevant information to further optimise the device and intervention before eventually designing an efficacy trial. Although the sample size is appropriate for a feasibility study [19], the study was not sufficiently powered to analyse the secondary outcome, efficacy. Moreover, we only analysed UE activity data of participants who adhered to the study. Therefore, those results should be interpreted with caution. Furthermore, our crossover design has some methodological concerns regarding secondary outcomes. Sub-acute stroke patients do not remain stable over time and show natural recovery [40]. By randomising and balancing the order of the interventions, we tried to limit the effects of patient-specific covariates on the outcomes in both groups. Due to the lower inclusion and randomisation based on twenty participants, we did not successfully balance the intervention order over the included participants. In addition, we could not evaluate whether these covariates were equally distributed at the start of intervention and in control conditions between the two orders because we only measured UE function, capacity and stroke severity at the baseline in our study. Further, we did not include a washout period. We did not expect carryover effects because the duration of the intervention was only two weeks, which is likely too short to result in changes in movement behaviour after stopping the intervention [16]. Despite this, we cannot be sure there is no remaining effect of the feedback on UE activity during the control period. To our knowledge, there are no recommendations regarding a washout period for crossover studies investigating movement behaviour interventions. Other crossover studies investigating such interventions did not include such a period [41] or concluded that a washout period was unlikely to negate the effects [42]. However, if a carryover effect exists, our current study underestimates the intervention effect compared to the control group.

Based on the promising results of this study on feasibility and estimates of the effect size of an intervention with the AAT, we recommend proceeding with further research to evaluate the (long-term) efficacy of the AAT. The results of this study can be used to efficiently plan a randomised controlled trial with sufficient power. In addition, we recommend selecting patients who can potentially benefit from this intervention based on upper extremity recovery profiles for stroke patients [43,44]. Our study showed that the most reported reason for dropout was participants’ experience of psychological burden when wearing the device in combination with other therapy during rehabilitation admission. We suggest applying the AAT as a home-based intervention after discharge from rehabilitation should be considered because the literature indicates that home-based rehabilitation may, depending on the rehabilitation phase and participants’ characteristics, further improve UE function and capacity or prevent patients from deterioration [45,46].

## 5. Conclusions

This study evaluated the feasibility and explored the efficacy of the Arm Activity Tracker to stimulate upper extremity activity in stroke patients. We found adequate acceptance and usability and increased paretic UE activity in participants who adhered to the study. The dropout rate was notable, and the reasons for dropout should be further evaluated. A sufficiently powered trial should be performed to analyse efficacy, including a follow-up to measure long-term outcomes.

## Figures and Tables

**Figure 1 sensors-23-05868-f001:**
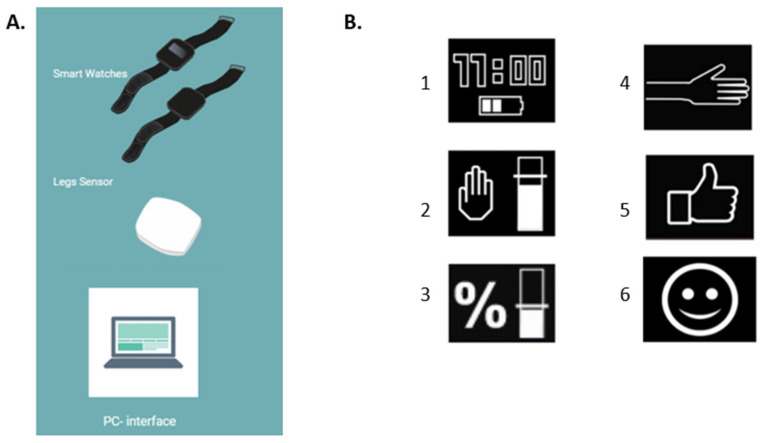
The components (**A**) and the visual feedback (**B**) of the Arm Activity Tracker. (**A**): The smartwatches for both wrists, the Activ8 sensor for the thigh and the computer interface for therapists. (**B**): The smartwatch display showing (1) the home screen; (2) the pre-set paretic UE activity goal; (3) the pre-set UE activity ratio goal; (4) indication to the user to increase UE activity; (5) indication the user is on track to reach a goal; (6) indication the user has reached a goal.

**Figure 2 sensors-23-05868-f002:**
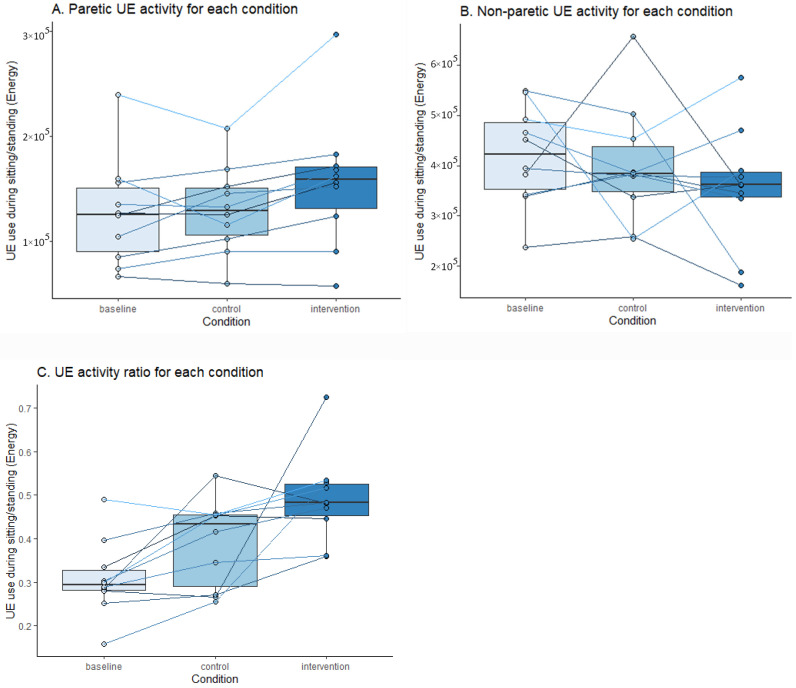
Paretic, non-paretic UE activity and ratio during each condition. Figure 2 shows (**A**) the average paretic activity, (**B**) non-paretic UE activity and (**C**) the ratio in energy during baseline, control and intervention conditions. The box plots show the median UE activity on a group level; the lines indicate the median UE activity for individual participants.

**Table 1 sensors-23-05868-t001:** Baseline characteristics of the participants.

	All Participants (n = 17)	Intervention Condition First (n = 10)	Control Condition First (n = 7)
Age, years *	61 (51–64)	61.5 (53–64)	55 (35–66)
Type of stroke, n Ischemic/hemorrargic (%)	17 (100%)/0	10 (100%)/0	7 (100%)/0
Time since stroke, days *	33 (25–60)	33 (28–60)	27 (18–84)
Gender male, n (%)	10 (59%)	4 (40%)	6 (85.7%)
NIHSS, score (range: 0–42) *	4 (2.5–7)	4 (2–5)	5.5 (3–8)
FMA UE, score (range: 0–66) *	43 (35–54)	50(41.5–62)	51 (35–60)
ARAT, score (range: 0–57) *	38 (14–49)	31 (6–49)	38 (14–57)
MI UE, score (range: 0–100) *	72 (61–83)	76 (39–83)	70 (66–85)
ATI, score (range: 1–6) *	4.83 (3.75–5.85)	4.78 (3.56–5.56)	4.89 (4.11–5.67)

* Values are presented as median (IQR). NIHSS: National Institutes of Health Stroke Scale, FMA: Fugl-Meyer assessment, ARAT: arm reach activity test, MI: motricity index, ATI: affinity for technology interaction scale.

**Table 2 sensors-23-05868-t002:** Technology acceptance model scores.

	Item TAM [27]	Median (IQR)
**Perceived usability**	1. Easy to use	7 (6–7)
	2. Easy to learn how to use	7 (6–7)
	3. Clear and easy to understand how to use	7 (6–7)
	4. Messages are clear	6 (5–7)
**Perceived usefulness**	5. Stimulates arm use	6.5 (4–7)
	6. Provides insights on arm use	6.5 (4.5–7)
	7. Usefulness	6 (4.5–7)
	8. Improves my arm rehabilitation	6 (4–6.5)
**Attitude towards use**	9. Would like to use	5 (3.5–6)
	10. Good to use it for my recovery	5.5 (4–7)
	11. Would like to use it at home	3.5 (1.5–6)
	12. My family and friends would support the use	7 (6.5–7)
**Intention to use**	13. Intention to use when it is available	4.5 (1–6)
	14. Intention to use it often	4.5 (1.5–6)
	15. Would use it when needed for my rehabilitation	6 (6–7)
	16. Intention to use it at home	4.5 (3–6)
	**Total score (Range: 16–112)**	94 (77–111)

**Table 3 sensors-23-05868-t003:** System usability scale scores.

Item SUS [31]	Median (IQR)
**1.Use it frequently**	3.5 (3–4)
**2. Unnecessary complex ***	1 (1–2)
**3. Easy to use**	4.5 (4–5)
**4. Need support of a technical person ***	1 (1–1.5)
**5. Functions are well integrated**	3.5 (3–4)
**6. Too much inconsistency ***	1.5 (1–2.5)
**7. Learn to use it very quickly**	4 (4–4.5)
**8. Very cumbersome to use ***	1 (1–2)
**9. Very confident in using**	4 (3–4.5)
**10. Need to learn a lot to use ***	1 (1–1)
**Total (Range: 0–100) ****	77.5 (75–78.5)

* Even numbered questions are negative questions, with a score of 1 indicating an excellent score. ** To calculate the SUS total score, we converted the original scores to a 0–100 scale.

## Data Availability

The data presented in this study are available on request from the corresponding author. The data are not publicly available due to privacy/ethical restrictions.

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
