# Peer review of "A Sensor-Based Feedback Device Stimulating Daily Life Upper Extremity Activity in Stroke Patients: A Feasibility Study"

_sensors, 2023, doi:10.3390/s23135868_

Round 1

Reviewer 1 Report

1. The number of experimental objects in the article is small, and it is suggested that more experimental objects should be provided to verify the accuracy of the investigation in the article.

2. The title of the article is not eye-catching enough. It is recommended that the author set the title hierarchy.

3. The fourth page of the article has most of the blank space. It is recommended that the author rearrange the content.

4. There is an issue with the header format of table1 on page 7 of the article, and the author needs to rearrange it.

Author Response

  1. The number of experimental objects in the article is small, and it is suggested that more experimental objects should be provided to verify the accuracy of the investigation in the article.

Response: Our sample size was based on our primary research aim: evaluating feasibility. We agree that a larger sample size would have been more optimal for evaluating feasibility. As described in our manuscript (Results, lines 247-248), due to COVID restrictions we could not continue our inclusion of up to 20 participants, but we were able to include more than the required minimum number of 12 participants.

We agree for our secondary outcome, exploring efficacy, the sample is small even in a cross-over design. This has been acknowledged in our Discussion (lines 395-396). Based on your important comment, we further explained our reasoning for the sample size in the text by making the following adjustments:

Materials and methods (lines 87-88): “Based on ambiguous literature about sample size justification for feasibility studies, we aimed to recruit a minimum number of 12 [20] and a preferred number of 20 participants [21], from Rijndam Rehabilitation Centre (Rotterdam, the Netherlands) during inpatient rehabilitation.”

Discussion (lines 388-395): “…However, our feasibility study was pragmatically designed to evaluate whether an intervention study can be done instead of performing a miniature version of a main trial [41]. The importance of such work has been recognised as an essential step before conducting clinical trials with large numbers of participants and a longer follow-up [41]. Therefore, our study design, including a shorter intervention period, fits the aims of a feasibility study and provides relevant information to further optimise the device and intervention before eventually designing an efficacy trial.”

  1. The title of the article is not eye-catching enough. It is recommended that the author set the title hierarchy.

Response: Thank you for your suggestions. We agree and therefore changed the title to a more eye-catching and informative title.

Title: A sensor-based feedback device stimulating daily life upper extremity activity in stroke patients: a feasibility study.

  1. The fourth page of the article has most of the blank space. It is recommended that the author rearrange the content.

Response: Thank you for noticing. We have contacted the editorial office since they formatted our version manuscript into the Sensors format. By doing this the position and outline of some tables and figures changed. The editorial office advised us to make adjustments to our own version of the manuscript (not the Sensors format). Before processing the article in the Sensors format, proofs will be sent and the outline of tables and figures can be adjusted at that stage.

  1. There is an issue with the header format of Table 1 on page 7 of the article, and the author needs to rearrange it.

Response: Thank you. We will explicitly pay attention to it when the article is further processed in Sensors format. 

Reviewer 2 Report

The aim of this article is to assess the feasibility of using the Arm Activity Tracker (AAT) as a device to stimulate activity in the paretic upper extremity of stroke patients. The subject matter is interesting and novel. I would like to make some comments to the authors and ask them some questions about their work.

The introduction is clear and supported by an adequate bibliography.

The material and methods section raises some questions for me:

- What is the basis for an intervention of only 2 weeks, and is it based on clinical evidence?

- Why has washout period not been included when it is the methodological norm in this type of crossover study?

- Why do you use references 20 and 21 when recruiting participants for the study? (page 2, line 88). Number 20 is linked to pharmacologically based clinical studies, while later, in the Discussion, instead of citing number 21, number 38 is cited. Could you explain this to me?

- Is the TAM questionnaire validated in Dutch, and has it been used in previous studies in Dutch?

- Could you tell me what are the "predefined cut-off points" you have used (page 6, lines 215-216)?

In addition, I would suggest, in this section, that Figure 1 be moved to the previous page and have only one explanatory caption, not two (line 160 and lines 163-166), which should go after the figure, not before as in the case of tables (the same with Figure 2, page 10).

As for the results, I suggest that:

- The explanatory footnote should not remain on a different page from the accompanying table (Table 2).

- The items linked to each study factor in Table 2 should be clearer.

- Each item of the TAM questionnaire and the SUS scale should occupy a single row.

- What cut-off points were taken to talk about small, medium or large effect sizes? (page 9, lines 296-297 and lines 301-302).

In relation to the Discussion, I think that the importance should be given to the sample loss of the study in relation to the interpretation of the results. The loss is important enough not to think that it could have a huge impact on how data are interpreted.

Despite this loss and despite pointing out the lack of statistical power of the study to be able to analyse the effects of the intervention (page 11, line 348), as well as the numerous limitations of the study (page 12, lines 372-389), the authors insist on highlighting the effect size of the intervention performed (page 11, line 350) and conclude that there is good acceptability and usability of the AAT device, as well as an increase in paretic upper limb activity (page 12, lines 404-405). Isn’t their interpretation too optimistic?

It could also be a limitation that only 2 weeks of intervention were performed (the other 3 are to establish the baseline activity level of the upper limb and the control period).

Finally, it should be noted that the bibliography does not follow the indications of the journal and data are missing in references 5, 20, 22 or 25. Possibly, it would be more appropriate to replace reference 29 with this one: Brooke, J.B. SUS-A quick and dirty usability scale. Usability Eval. Ind. 1995, 189, 4-7. doi: 10.1201/9781498710411-35

Author Response

The material and methods section raises some questions for me:

  • What is the basis for an intervention of only 2 weeks, and is it based on clinical evidence?

Response: Thank you for your comment. We carefully considered the duration of the intervention for the aims of this feasibility study. We clarified this by making the following additions to the manuscript:

Discussion (lines 385-395): “The intervention period in the study was two weeks, which is short compared to most interventions to change stroke patients’ movement behaviour [16] or UE capacity [10]. As we suggested, the duration of the intervention period could have influenced the results found in this study e.g., the dropout rate and the effect sizes. However, our feasibility study was pragmatically designed to evaluate whether an intervention study can be done instead of performing a miniature version of a main trial [41]. The importance of such work has been recognised as an essential step before conducting clinical trials with large numbers of participants and a longer follow-up [41]. Therefore, our study design, including a shorter intervention period, fits the aims of a feasibility study and provides relevant information to further optimise the device and intervention before eventually designing an efficacy trial.”

  • Why has washout period not been included when it is the methodological norm in this type of crossover study?

Response: Thank you for this comment. We clarified our assumptions about carry-over effects in the text.

Discussion (lines 409-414): “We did not expect carry-over effects since the duration of the intervention was only two weeks, which is likely too short to result in changes in movement behaviour after stopping the intervention [16]. Despite this, we cannot be sure there is no remaining effect of the feedback on UE activity during the control period. In case a carry-over effect exists, our current study underestimates the intervention effect compared to the control group.

  • Why do you use references 20 and 21 when recruiting participants for the study? (page 2, line 88). Number 20 is linked to pharmacologically based clinical studies, while later, in the Discussion, instead of citing number 21, number 38 is cited. Could you explain this to me?

Response: Thank you for this comment. The sample size for feasibility studies is not strictly determined, the literature suggests different sample sizes. Based on widely accepted literature we made our decisions regarding sample size. To clarify how on and on which literature we based our sample size, we made the following changes to the manuscript:

Materials and methods (lines 87-88): “Based on ambiguous literature about sample size justification for feasibility studies, we aimed to recruit a minimum number of 12 [20] and a preferred number of 20 participants [21], from Rijndam Rehabilitation Centre (Rotterdam, the Netherlands) during inpatient rehabilitation.”

Discussion (lines 388-395): “However, our feasibility study was pragmatically designed to evaluate whether an intervention study can be done instead of performing a miniature version of a main trial [41]. The importance of such work has been recognised as an essential step before conducting clinical trials with large numbers of participants and a longer follow-up [41]. Therefore, our study design, including a shorter intervention period, fits the aims of a feasibility study and provides relevant information to further optimise the device and intervention before eventually designing an efficacy trial.”

  • Is the TAM questionnaire validated in Dutch, and has it been used in previous studies in Dutch?

Response: Thank you for this comment. It has been validated in an English-speaking population (Methods, line 192). We added the following to the manuscript:

Materials and Methods (lines 193-194): “The TAM has been used in Dutch-speaking populations [30]. In this study, we also used the TAM in Dutch.”

  • Could you tell me what are the "predefined cut-off points" you have used (page 6, lines 215-216)?

Response: Thank you for noticing this was not clear. We were referring to this sentence in the Methods (line 203) “SUS score above 68 can be considered adequate usability [29].” We made the following adjustment to clarify:

Materials and Methods (lines 223-224): we removed this sentence because it did not apply to the TAM: “We compared those to the pre-defined cut-off scores.”

Results (lines 277-278): “The twelve participants who completed the study protocol scored above the pre-set score of 68 on the SUS (median 77.5; IQR 75.0-87.5), indicating adequate usability [29].”

  • In addition, I would suggest, in this section, that Figure 1 be moved to the previous page and have only one explanatory caption, not two (line 160 and lines 163-166), which should go after the figure, not before as in the case of tables (the same with Figure 2, page 10).

Response: Thank you for noticing. We placed the captions of the figures below the figures. In addition, we have contacted the editorial office, since they formatted our version manuscript into the Sensors format. By changing the format, tables and figures did not fit on the pages. The editorial office advised us to make adjustments in our version of the manuscript (not Sensors format). Before processing the article in the Sensors format, proofs will be sent and the outline of tables and figures can be adjusted at that stage.

As for the results, I suggest that:

  • The explanatory footnote should not remain on a different page from the accompanying table (Table 2).

Response: Please, see our response to your previous comment.

  • The items linked to each study factor in Table 2 should be clearer.

Response: According to your advice, we changed the layout of Table 2. We hope this clarifies which items are linked to each of the studied factors in Table 2.

  • Each item of the TAM questionnaire and the SUS scale should occupy a single row.

Response: Thank you for noticing, we will explicitly pay attention to it when the article is further processed in Sensors format.

  • What cut-off points were taken to talk about small, medium or large effect sizes? (page 9, lines 296-297 and lines 301-302).

Response: To clarify the cut-off points we adjusted the following sentence:

Materials and methods (lines 240-241): “Generalised eta squared (η2G) with Confidence Intervals (CI) were calculated to identify effect sizes of the intervention. Effect sizes were deemed small (0.02), medium (0.13) or large (0.26) [34].”

  • In relation to the Discussion, I think that the importance should be given to the sample loss of the study in relation to the interpretation of the results. The loss is important enough not to think that it could have a huge impact on how data are interpreted.

Despite this loss and despite pointing out the lack of statistical power of the study to be able to analyse the effects of the intervention (page 11, line 348), as well as the numerous limitations of the study (page 12, lines 372-389), the authors insist on highlighting the effect size of the intervention performed (page 11, line 350) and conclude that there is good acceptability and usability of the AAT device, as well as an increase in paretic upper limb activity (page 12, lines 404-405). Isn’t their interpretation too optimistic?

Response: Thank you for highlighting this point. We agree that the sample loss has an important effect on the results. Adherence is one of the measures for our primary outcome, feasibility. In our manuscript, we explicitly mentioned that results about acceptability and usability were found in participants who adhered to the study and that drop-outs should be further investigated regarding adherence (Abstract lines 27-28,30; Results lines 270,277,290; Discussion line 332; Conclusion lines 428-430).

Regarding the secondary outcome, we advise interpreting the results with caution as described in the Discussion (lines 397-398). To make this more explicit we added the following sentence:

Discussion (lines 396-397): “Moreover, we only analysed UE activity data of participants who adhered to the study.”

  • It could also be a limitation that only 2 weeks of intervention were performed (the other 3 are to establish the baseline activity level of the upper limb and the control period).

Response: Thank you for your comment. We agree that this could be a limitation, but we decided that for this study a two-week intervention was appropriate for the aims of a feasibility study. We clarified this by making the following additions to the Discussion (lines…) of our manuscript. Please, see also our reply to your comments 1 and 3.

  • Finally, it should be noted that the bibliographydoes not follow the indications of the journal and data are missing in references 5, 20, 22 or 25. Possibly, it would be more appropriate to replace reference 29 with this one: Brooke, J.B. SUS-A quick and dirty usability scale. Usability Eval. Ind. 1995, 189, 4-7. doi: 10.1201/9781498710411-35

Response: Thank you for noticing. We have adjusted the references you pointed out in line with the journal requirements.

Reviewer 3 Report

The manuscript discloses the evaluation of an Arm Activity Tracker based on a network of three accelerometers, in stroke patients.

The study clearly showed good acceptance and ease of use of the device. The tracker also appears to stimulate activity in the paretic upper limb, but the study was not designed to objectively assess the performance of the device as disclosed by the authors.

The authors should probably comment further on the drop out. In particular, is the psychological burden on patients strong enough not to accept the use of the tracker? Can the design of the tracker be improved to increase adoption? Were patients asked questions to explain why they refused the tracker? These points should most likely be addressed.

The robustness of monitoring data should also be documented. In particular, how sensitive is the location of the leg sensor in terms of generating data and identifying movement patterns and sitting/standing posture?

It would be nice to have a photo of the position of the leg sensor.

Even if the study was not powered for performance evaluation, how might the placebo effect influence the study results? It seems that the Control conditions may have some impact, even if it is not significant in the series of patients included. The order of the intervention and control conditions should also be taken into account.

Check minor spellings

Author Response

1) The authors should probably comment further on the dropout. In particular, is the psychological burden on patients strong enough not to accept the use of the tracker? Can the design of the tracker be improved to increase adoption? Were patients asked questions to explain why they refused the tracker? These points should most likely be addressed.

Response: Thank you for your comment. We agree that the dropout rate and the reasons for dropout are important findings of this feasibility study. We also acknowledge further investigation of this particular reason for drop-out is important to optimize the design of the device before conducting a larger trial with this device (Discussion lines 343-352). However, in our current study, we do not have more detailed data on this. To clarify what we did ask regarding the dropouts, we made the following adjustment:

Materials and methods (187-188): “The trained physiotherapist asked and reported participants’ reasons for dropping out during the baseline, intervention and control conditions.”

2) The robustness of monitoring data should also be documented. In particular, how sensitive is the location of the leg sensor in terms of generating data and identifying movement patterns and sitting/standing posture?

Response: Thank you for pointing this out. We agree this is relevant information for the reader. We, therefore, made the following adjustments to the manuscript:

Materials and methods (lines 127-133):The AAT consists of three small, lightweight accelerometer sensors (Figure 1A): one worn on each wrist and one attached to the front of the non-paretic thigh [22]. The AAT accurately measures paretic UE activity in stroke patients (overall agreement: 75%) by recording arm movements only during sitting and standing periods to correct for involuntary UE movement caused by whole-body movements such as walking [18, 22]. The leg sensor accurately detects whole-body movements and postures in stroke patients such as walking, sitting, and standing (accuracy ranging from 82 to 100%) [23].

3) It would be nice to have a photo of the position of the leg sensor.

Response: Please, review our response to your previous comment. Reference 22 refers to the specific locations of the sensors. Besides, we made the following adjustment to specify the location:

Materials and methods (lines 127-128): “….one worn on each wrist and one attached to the front of the non-paretic thigh [22].”

4) Even if the study was not powered for performance evaluation, how might the placebo effect influence the study results? It seems that the Control conditions may have some impact, even if it is not significant in the series of patients included. The order of the intervention and control conditions should also be taken into account.

Response: Thank you for your comment. We agree it is important to consider potential placebo effects in studies evaluating efficacy. However, since we evaluate the effect of feedback based on the AAT compared to no feedback (as described in Materials and Methods: Intervention), we feel the placebo effect of wearing an activity tracker it is not that relevant to this study. If we aimed to estimate the effect of the activity tracker + feedback compared to no activity tracker and no feedback, the placebo effect might influence the results and would have been relevant to discuss. Effects of the order of the intervention condition were taken into account in the statistical analysis (Materials & Methods: Statistical analysis).

Although we feel it is not directly relevant to our comparison, we are willing to add it to our manuscript if you still advise us to add this to the manuscript.

5) Comments on the Quality of English Language

Check minor spellings

Response: We checked the manuscript on spelling and adjusted this when needed.

Round 2

Reviewer 1 Report

1、The title of the article is not eye-catching enough. It is recommended to set the title hierarchy, which has not been changed.

2、The formatting of the article still needs to be further modified and improved, and the page should not have a lot of blank space as much as possible.

Author Response

Thank you for your comments and suggestions. Below, you can find our responses. 

1、The title of the article is not eye-catching enough. It is recommended to set the title hierarchy, which has not been changed.

Response: Thank you again for this comment.

We asked the assistant editor to contact you about a further explanation of setting the title hierarchy. We now changed the title hierarchy according to our understanding of your comment. Please, let us know if this is not what you suggested with setting the title hierarchy.

Title: “A sensor-based feedback device stimulating daily life upper extremity activity in stroke patients: a feasibility study.”

2、The formatting of the article still needs to be further modified and improved, and the page should not have a lot of blank space as much as possible.

Response: Thank you for your comments on the formatting of the manuscript. We contacted the editorial office about this. Our article will be formatted into the format of Sensors, and when that is done, we can review the formatting of tables and figures, and avoid white space. Since we did not provide our manuscript in sensor format, it is not necessary to change the formatting in our Word version before the editorial office has formatted it into Sensors format. Thank you for your understanding, we will make sure to check the formatting in the article proofs before the article is published.

Reviewer 2 Report

In this second revision of this paper, I did not have access to the new version. The new version of the article that I was able to download from the MDPI platform was the same as the one initially reviewed. This means that I do not see the changes that the authors point out in their response. I take their word for it, but I cannot confirm that they have made the changes they say that they have made. 

In general, the authors have answered my doubts and considerations. I still think there are certain weaknesses, such as: 

-  In relation to the authors' response about a 2-week intervention and no washout period:  

I understand and share that it is a short intervention and a washout period might not be necessary, but the title and aim of the paper points out that it also wants to explore the efficacy of the sensor, not just its feasibility. Therefore, the design could condition the results. Moreover, the title of the article could refer only to feasibility. 

- In relation to the justification of references 20 and 21: 

The sample size in a feasibility study is not strictly determined, but should be based on similar work. It may not be appropriate to base a non-pharmacological therapy work on a pharmacological therapy reference. 

- In relation to the use of the TAM in Dutch: 

As it is not validated in Dutch, its use in Dutch should at least be referenced in previous studies. 

- Regarding the importance of dropouts in the discussion: 

It seems to me that assessing feasibility only on the basis of who has participated is a partial view in the assessment, and a weakness in the justification of that feasibility. 

I am therefore hesitant to recommend the publication of this work. If the journal sees fit to publish it, I will not oppose it either.

Author Response

Thank you again for your comments. We are sorry to hear you were not able to review our adjusted manuscript. We will again upload our adjusted version and hope you are able to see this version this time. Below, you can find our responses to your comments.

1_ In relation to the authors' response about a 2-week intervention and no washout period:  

I understand and share that it is a short intervention and a washout period might not be necessary, but the title and aim of the paper points out that it also wants to explore the efficacy of the sensor, not just its feasibility. Therefore, the design could condition the results. Moreover, the title of the article could refer only to feasibility. 

Response: Thank you for your comment. We agree that the title can be confusing since it does not clearly shows the hierarchy between the primary and secondary outcomes of this study. Another reviewer already commented on the title in the previous review. We therefore already changed the title. Unfortunately, you were not able to see this since you could not download the previous adjusted version. We expect that with this change, we also answered your comment.

Title: “A sensor-based feedback device stimulating daily life upper extremity activity in stroke patients: a feasibility study.”

2) In relation to the justification of references 20 and 21: 

The sample size in a feasibility study is not strictly determined but should be based on similar work. It may not be appropriate to base a non-pharmacological therapy work on a pharmacological therapy reference. 

Response: Thank you again for this comment. In addition, we now refer to another reference from a similar study in ten stroke patients to evaluate the usability and acceptance of a feedback device, published in a peer-reviewed Q1 journal:

Methods: “Based on ambiguous literature about sample size justification for feasibility studies and similar work, we aimed to recruit a minimum number of 12 [20,21] and a preferred number of 20 participants [22].”

21: Held JP, Klaassen B, van Beijnum BF, et al. Usability Evaluation of a VibroTactile Feedback System in Stroke Subjects. Front Bioeng Biotechnol. 2016;4:98.

3) In relation to the use of the TAM in Dutch: 

As it is not validated in Dutch, its use in Dutch should at least be referenced in previous studies. 

Response: To support the use of the Dutch version, we referred to another study using a Dutch version of the TAM.

Methods (lines 194-196): For the Dutch-speaking participants in this study, we translated the TAM into Dutch. The TAM has been used in Dutch-speaking populations before [31]. 

  1. Askari M, Klaver NS, van Gestel TJ, et al. Intention to use Medical Apps Among Older Adults in the Netherlands: Cross-Sectional Study. J Med Internet Res. 2020 Sep 4;22(9):e18080.

4) Regarding the importance of dropouts in the discussion: 

It seems to me that assessing feasibility only on the basis of who has participated is a partial view in the assessment and a weakness in the justification of that feasibility. 

I am therefore hesitant to recommend the publication of this work. If the journal sees fit to publish it, I will not oppose it either.

Response: We agree that adherence, measured as the dropout rate, is an important outcome of feasibility. In our study, it is one of the primary outcome measures of feasibility, together with acceptance and usability (Methods lines 184-206). In our conclusions, we feel we are transparent in our interpretation of the data. We conclude good acceptance and usability in participants who adhered (based on the SUS and TAM scores found in this study) and we conclude that reasons that require further research ((Conclusion lines 430-443). We have discussed the reasons for dropping out. We agree that these are important findings regarding the feasibility and therefore recommend further research into the perceived psychological burden as a reason for dropping out, and we suggested this might have to do with the timing of the intervention and we recommend considering the timing carefully when using this intervention (discussion lines 336-355).

We, therefore, believe our conclusions regarding the feasibility (consisting of adherence, usability, and acceptability) are validly based on our data. According to your comment, we emphasised the significant drop-out rate by adjusting our conclusion.

Conclusion: This study evaluated the feasibility and explored the efficacy of the Arm Activity Tracker to stimulate upper extremity activity in stroke patients. We found adequate acceptance and usability, and an increase in paretic UE activity in participants who adhered to the study. The dropout rate was notable, and the reasons for dropouts should be further evaluated. A sufficiently powered trial should be performed to analyse efficacy, including a follow-up to measure long-term outcomes.